# Components of Artificial Neural Networks Realized in CMOS Technology to be Used in Intelligent Sensors in Wireless Sensor Networks

**DOI:** 10.3390/s18124499

**Published:** 2018-12-19

**Authors:** Tomasz Talaśka

**Affiliations:** 1Faculty of Telecommunication, Computer Science and Electrical Engineering, UTP University of Science and Technology, 85-796 Bydgoszcz, Poland; talaska@utp.edu.pl; 2Aptiv Services Poland S.A., 30-399 Kraków, Poland

**Keywords:** wireless sensor networks, intelligent sensors, artificial neural networks, ASIC, CMOS technology, low power solutions, parallel data processing

## Abstract

The article presents novel hardware solutions for new intelligent sensors that can be used in wireless sensor networks (WSN). A substantial reduction of the amount of data sent by the sensor to the base station in the WSN may extend the possible sensor working time. Miniature integrated artificial neural networks (ANN) applied directly in the sensor can take over the analysis of data collected from the environment, thus reducing amount of data sent over the RF communication block. A prototype specialized chip with components of the ANN was designed in the CMOS 130 nm technology. An adaptation mechanism and a programmable multi-phase clock generator—components of the ANN—are described in more detail. Both simulation and measurement results of selected blocks are presented to demonstrate the correctness of the design.

## 1. Introduction

In typical wireless sensor networks, the role of particular sensors is to register a given signal from the environment, perform an initial data preprocessing and then to transfer it to a base station where an artificial neural network (ANN) may be used to carry out a more detailed analysis. Signal processing performed at the sensor level typically includes an anti-aliasing filtering, analog-to-digital conversion, and optionally data compression to reduce the amount of data transferred over the wireless network. One of the problems associated with WSN development is high energy required to transfer data. Radio-frequency (RF) communication block, used of this purpose, consumes even 80–90% of total energy consumed by the overall sensor [1,2].

To reduce the described problem, we propose a solution, in which to a much greater extent than now, data processing will take place directly at the sensor level. The proposed scenario is as follows. After the analog-to-digital conversion, a data preprocessing takes place inside the sensor to prepare a training file for an ANN integrated together with other sensor components in a single device. Thus, subsequent data analysis would be performed inside the sensor. In this situation, only the results provided by the ANN would be transferred to the base station, which usually means a much smaller amount of data than in the case of raw data.

An important question here is the one about the profitability of this approach. The problem can be defined, for example, as follows: Does the energy consumed by the ANN implemented in the sensor will be smaller than the energy saved by the radio-frequency (RF) transmission block in case of reduced operation time? This problem has been studied in more detail in previous works on hardware implemented ANNs, which may be found in the literature [3]. The answer to some extent depends on the type of the processed data, and thus the data rate. The investigation presented in [3] shows that optimized parallel ANN, working at data rates at the level of 1000–2000 samples/s dissipates the power in-between 2 and 30 μW, for 64 and 1000 neurons, respectively, for 10 inputs of the network. In many real applications, the number of neurons is closer to the lower value. Such data rates are sufficient, for example, while processing various biomedical data. Even at data rates two orders of magnitude higher, the ANN would dissipate power not exceeding 200–500 μW. For the comparison, the RF communication block may dissipate power up to 15 mW, as reported in [1,2].

For the described problem, however, we can look not only from the point of view of the energy consumption. When the ANN is implemented inside the wireless sensor, then in the overall WSN we have *N* distinct ANNs operating in parallel (distributed data processing), where *N* is the number of sensors in a given WSN. If data collected by all these sensor would have to be analyzed by the ANN implemented in a base station, a throughput problem may appear. Such ANN would have to be multiplexed and switched between data streams coming from particular sensors. An additional problem appears when there is no synchronization between particular sensors. In other words, the distributed data processing may be more convenient at all.

One of the problems here is how to implement the ANN, so that it features a small silicon area and very low energy consumption. Such sensors can be described as intelligent wireless sensors. A substantial limitation of data transmitted over the wireless network will, in turn, strongly reduce the energy consumed by the overall sensor. As a result, the sensor to higher degree would be able to operate with the energy scavenged from the environment. This approach may greatly simplify the assembly, as well as the operation of such devices over a longer period. Our investigations show that even large neural networks, implemented in the CMOS technology, operating at data rates of several thousand samples/s (sufficient in many situations) dissipates the power not exceeding a few μW [4].

In this work, we present a contribution to the described problem. For many years, we have performed investigations toward efficient transistor level implementations of very low power, parallel ANNs in which each neuron is represented by a separate circuit. In this paper, we present a prototype integrated circuit containing novel implementations of main components of such neural networks, as well as selected components of the overall wireless sensor. The proposed circuits were optimized for the application in parallel ANNs. To make this possible, their structure had to be substantially simplified, however not at the expense of a degradation of the behavior of the overall ANN. The fabricated chip was positively verified by means of laboratory measurements.

In this work, we focus in particular on the adaptation mechanism for such ANN, as well as on a multiphase programmable clock used to control the ANN. One of the goals was to obtain flexible solutions, which means, for example, the ease of changing the number of input signals, i.e., the size of learning patterns. In this approach, the presented clock controls the stage of determining the distance between vectors of neurons weights and learning patterns. It is then used in the process of adapting neuron weights. It is also used to control the analog-to-digital converter (ADC) and filters used at the data preprocessing stage. It is therefore a block of universal application, playing a significant role in the whole sensor.

### 1.1. State-of-the-Art Study

#### 1.1.1. Wireless Sensors

In this section, we briefly present the proposed concept of the intelligent wireless sensors. Our goal is not to completely change the structure of existing wireless sensors, but rather to expand their functionalities. A typical wireless sensor consists of several components listed below [5,6,7,8]:front-end sensor used to collect analog data from the environment;anti aliasing filter and filters used to remove the noise from the signal;ADC used to convert measured analog signal into its digital counterpart; andRF communication block.

The requirements as well as the structure of particular components of the sensors depend on the application for which they are designed. In an in-door environment, for example in some medical applications [9], the robustness on extreme conditions may be not so important as in the case of the sensors to be used in the out-door environment, e.g., in automotive applications [10]. On the other hand, in the case of medical application, a larger miniaturization and lower energy consumption is required.

The first element in the presented data processing chain is the front-end sensor used to collect data from the environment. Depending on spectrum of the input data, a low-pass analog filter may be required to avoid aliasing that may occur during the following operation, which is the analog-to-digital conversion. The signal after the ADC may be further processed in various ways in a data processing unit.

In the proposed concept of the intelligent sensors, we assume a direct analysis of the collected signals by the use of an ANN integrated together with other components of the sensor. To make this possible, we propose to extend the scope of functionality of the processing unit by adding to it an artificial neural network and a block responsible for creating a training set. The structure of such a sensor is shown in Figure 1. The aim of this approach was to move some part of signal processing tasks from the external base station to the sensor itself. As a result, the sensor would communicate less frequently with the base station and thus consume less energy.

Since the key block in the described concept is a miniature ANN, the state-of-the-art should be more related to the neural networks themselves, instead of the sensors.

#### 1.1.2. ANNs Realized at the Transistor Level

There are not many studies in the literature on the implementation of ANNs as specialized integrated circuits. The existing realizations can be divided into digital and analog. For analog solutions, only three comparable designs can be indicated, as reported in [4,11,12]. The solutions reported in [11,12] are mentioned here only as an illustration, as they were designed in much older technologies (CMOS 2 μm) that are not comparable to currently used technology. In [4] is reported a fully analog neural network designed by us earlier, in a similar technology (CMOS 180 nm) as the current prototype (CMOS 130 nm). This ANN consisted of four neurons. Each of them contained three calculation channels corresponding to three inputs of the ANN and thus three weights per each neuron. The ANN worked in the current mode. All main components were designed by us from scratch [13,14,15]. Design of such NNs is a complex task, as after the fabrication no changes are possible. The design process requires comprehensive simulations to check as many scenarios as possible [16] to make the fabricated chip a universal solution.

As regards state-of-the art digital solutions, on the one hand, solutions based on field programmable gate arrays (FPGA) can be indicated [17,18,19,20]. However, such solutions are not comparable with our works, which results from the fact that in this case the emphasis is put on different aspects. For example, in our solutions, the miniaturization of the circuit, as well as reduction of energy consumption, is of paramount feature, while, in the FPGA realization, the emphasis is put rather on the convenience and time spent on the design, while the two mentioned parameters are usually of second importance.

On the other hand, to some extent, selected circuits designed for the pattern recognition applications may be comparable with self-organizing ANNs [21,22,23]. In such circuits, some components have a similar structure. For example, the block used to calculate the distance between two vectors of signals and the block used to determine which internal pattern is the most similar to an input pattern are the same as in the self-organizing ANNs. However, in these solutions, the adaptation block, which is one of the subjects of the presented work, is not used. There is also no clock generator described in these works.

### 1.2. Technical Background

In this work, we focus on self-organizing ANNs that allow extending the range of capabilities of existing wireless sensors and make them smarter. Such types of ANNs are used, for example, in the analysis of air pollutions to predict their levels in some periods. In the case of the proposed hardware implementation, such ANNs may be applied directly in wireless sensors in smart cities. Another group of applications includes the analysis of biomedical data, for example: ECG (Electrocardiography) and EMG (Electromyography) signals. In this case, the miniaturization of such ANNs may lead to efficient wireless body area networks used in medical health care.

Self-organizing ANNs feature relatively simple structure, which is suitable for low power and low chip area hardware realizations, as shown in our previous works [3,4,24]. This group of ANNs include the following learning algorithms: winner takes all (WTA), winner takes most (WTM) and the neural gas (NG). At most of the computation stages, these algorithms perform similar operations, as shown in Figure 2. Below are presented the subsequent steps carried out for each new input learning pattern *X*:Initialize the neuron weights that aims at distribution of the neurons over the overall input data space.Provide a new learning pattern *X* to the inputs of all neurons in the ANN (a data normalization may be required before following stages).Calculate distances ($d_{1}$, $d_{2}$, $d_{3}$, $d_{j}$, …, $d_{N}$) between the pattern *X* and the weight vectors *W* of all neurons in the ANN (*N* is the total number of neurons in the NN). The distance may be computed according to one of typical distance measures, for example the Manhattan (L1 norm) or the Euclidean (L2 norm) ones [3].Determine the neuron that is located in the closest proximity to a given pattern *X*. This neuron becomes the winner. Mathematically, the min($d_{1}$, $d_{2}$, $d_{3}$, $d_{j}$, …, $d_{N}$) operation is used at this stage.Determine the neighborhood of the winning neuron (explained in more detail below).Adapt the weights of the winning neuron and of its neighbors (in the WTM and the NG algorithms).

The adaptation of the neuron weights is performed according to the formula below, which is similar for all learning algorithms mentioned above:(1)$$W_{j}\left(k+1\right)=W_{j}\left(k\right)+\eta \left(k\right)\cdot G\left(\right)\cdot \left[X\left(k\right)-W_{j}\left(k\right)\right]$$

In this formula, the $W_{j}\left(k\right)$ is the weights vector of a *j*th neuron, in *k* cycles (iteration) of the learning process of the ANN, while η is the learning rate that determines the intensity of the learning process. The neurons that belong to the winner’s neighborhood, are trained with the intensities determined by the applied neighborhood function (NF) G() [3,24]. The values of the NF for particular neurons depend on distances between these neurons and the winning neuron.

The adaptation block is one of the most important components of the neural network. It computes an update of particular neuron weights, which are calculated as follows:(2)$$\Delta w_{i,j}\left(k\right)=\eta \left(k\right)\cdot G\left(\right)\cdot \left[x_{i}\left(k\right)-w_{i,j}\left(k\right)\right]$$

Before the learning process, the values of both the η(k) factor and the G() function are set to be in the range between 0 and 1, usually closer to 1. Then, during the learning process, these parameters are gradually reduced to zero. For η(k)=0, the adaptation process becomes inactive, and the NN enters the recall phase, in which the values of the weights are then constant. Thanks to setting the η(k) parameter to be less than 1, the subsequent learning process is convergent. At the beginning of the learning process, when the η and the G() values are relatively large, the modifications in weight values are greater than at the end of this process. In real data processing, however, when the input dataset suddenly changes, the η value should be enlarged again. In the case of hardware realizations of the neural networks, the changing rate of the η parameter should be additionally matched to various physical phenomena, especially visible in the case of analog solutions. In such networks, one can observe, for example, the leakage phenomenon of information from analog memory cells [13]. This problem may be solved, to some extent, by remaining the η coefficient at a non-zero level even in the recall phase. Under certain conditions, this allows compensating the the lost of the information. This problem, however, does not occur in digital networks that are the subject of this work.

The main differences between described three self-organizing ANNs are visible in the structure of the block responsible for the neighborhood mechanism. In the WTA algorithm, the neighborhood is not used. In this case, however, the so-called conscience mechanism may be added to allow for a stimulation of all neurons in the ANN [4]. In the WTM ANN, particular neurons are permanently linked to their neighbors, which means that distance is determined in the so-called map space and is independent on the distance in the input data space. In contrast, in the NG algorithms, the neighborhood is created “ad hoc” based directly on the current positions of particular neurons in the input data space. This requires sorting the neurons according to their distances to a given learning pattern *X* (sort($d_{1}$, $d_{2}$, $d_{3}$, $d_{j}$, …, $d_{N}$)).

In the WTM and the NG approaches, a neighborhood function (NF) is required to differentiate the strength with which the weights of particular neurons are adapted. Theoretical and simulation studies reported in [25] have shown that a triangular NF is a good approximation of the Gaussian one, while it requires much simpler hardware. Using this function allows for a substantial simplification of the overall adaptation mechanism.

One of the important parameters of the presented NNs is the measure of the distance between successive learning patterns *X* and vectors of neuron weights of particular neurons. Figure 2 shows two popular ways to calculate the distance. From the point of view of the learning process of the neural network, both norms allow obtaining comparable learning results, as shown in our previous works in this area [3,15]. However, the application of the L1 norm is of great importance from the point of view of the computational complexity. This in turn is very important in hardware realizations, in which it directly translates into the circuit complexity. In the case of the L1 norm, only simple operations of adding, subtracting and calculating the function returning the absolute value of abs() are needed. In the case of the L2 (Euclidean) approach, it is also necessary to carry out squaring and square root operations. The two operations in the case of the software realizations are not problematic, but, when implementing at the transistor level, this issue is important.

The results presented in this work are universal and can be combined with each of the two norms, as the distances described above are computed at a different (former) stage of the overall learning process of the ANN.

In the literature dealing with the ANNs, an important factor is the learning time or, in other words, the convergence time of the learning process. This time depends on various factors. These include the number of the learning patterns, the number of the inputs of the ANN, the number of neurons in all layers of the network, the applied learning algorithm, the values of particular parameters of the adaptation mechanism (η and G() mentioned above), the method of determining the learning error that is responsible for the appropriate correction of the weights, etc.

In general, ANNs may be divided into those trained with the so-called teacher and those without the teacher. The ANNs belonging to the first group usually feature more complex learning algorithm, which translates into a longer convergence time. We focus in this work on the second (self-organizing) group, as they feature less computation complexity, and thus a simpler hardware structure. In our former investigations, we observed that in the case of such ANNs, assuming a proper configuration, the learning process was convergent for a wide range of the input datasets. Some problems with the convergence were observed in the case of the WTA algorithms with an incorrect initialization [4].

## 2. Materials and Methods

### 2.1. Materials

In this section, we briefly present tools and materials used to design and verify the presented prototype chip. First, we briefly describe the chip and its design process. Then, we provide a description of the measurement setup. The realized chip has been verified by the use of a custom designed PC board with on-board programmable devices such as XC9572XL (Xilinx, San Jose, CA, USA). The board closely cooperates with MyRio card, equipped with Xilinx Z-7010 device that contains two ARM Cortex-A9 cores, a set of programmable gates and I/O circuits that are equivalent to the Artix-7 FPGA. It is coupled with 256 MB RAM and 512 MB Flash memories. These devices allow programing the prototype chip and facilitating a full range of tests. Selected results are presented in next Section.

#### 2.1.1. Chip Design—Cadence Environment

The prototype chip was designed in the CMOS 130 nm technology, in the professional Cadence environment. Virtuoso tool was used for layout design. Spectre simulator as well as the LVS (layout-vs.-schematics) module were used for the chip verification. Before the fabrication, the chip was thoroughly verified by means of corner analysis, in which simulations were carried out for different values of the process, voltage and temperature conditions. The temperature varied between −40 and +140 °C, and the supply voltages between 0.8 and 1.2 V. The simulations were carried out for standard, slow and fast transistor models. In Figure 3, we present layout of one of the most important block—a programmable 10-phase clock generator, used in different parts of the sensor, and at different states of the learning process of the ANN.

The realized chip is a reconfigurable/programmable device. This means that the parameters of particular components as well as the connection scheme in the overall chip may be easily reconfigured. The problem we faced was a limited number of external pins, thus particular digital inputs and outputs had to be multiplexed in a proper way. This required a development a programmable I/O block, composed of configuration switches, an address decoder and the memory block. The overall chip area including the external pads equals 1.4 mm${}^{2}$, with the areas of particular developed components not exceeding 0.06 mm${}^{2}$.

#### 2.1.2. Measurement Setup

The main element of the measurement setup is the PCB board with the socket used to simplify the mounting procedure. The board also includes a dedicated power supply system, a digital I/O connector, a cross-over matrix in the form of a CPLD circuit, and I/V and V/I converters and buffers for analog lines.

The PCB is equipped with an appropriate measuring equipment. As a front-end measurement device, we used MyRio 1950 measuring card that operates under the control of the LabView (LV) environment. The card by using a built-in FPGA and the FIFO DMA registers allows exchanging digital measurement data with the tested device, with a step of 25 ns. It is also used to trigger other measurement components, thus allowing for a joint operation of different devices such as oscilloscopes and generators, while maintaining a common “timeline”. Immediately before the measurement, the LV configures the devices and fills the buffers. After the measurement, data obtained from the devices are collected, unified, adapted to the common timeline and saved to a csv file, along with the headers. Such files may be immediately further processed using such tools as, for example, dataplot or Microsoft Excel. The measuring setup is additionally equipped with a three-output programmable, precise DC power supply, and, depending on the type of measurement, also with Textronix oscilloscopes and generators. However, it should be remembered that the type and quantity of equipment required at the measurement station changes and is adapted on an ongoing basis depending on the type of measurement. The measuring setup is shown in Figure 4.

### 2.2. Methods—Solutions for the Adaptation Mechanism and Clock Generator

The proposed digital ANN operates in a mixed synchronous-asynchronous mode. This means that part of the operations is performed sequentially, but asynchronous processing is introduced wherever possible, which allows for significant simplification of the structure of the used circuits. The network works with multi-bit signals, which means that each input datum $x_{i}$, as well as the corresponding neuron weight $w_{i,j}$, is a multi-bit number (usually 16-bit) [26]. Lower resolutions, however, may be used to test the proposed solution. Providing, in parallel, a full learning pattern into the network ($X=\{x_{1},x_{2},\ldots ,x_{n}\}$) is impractical, considering various possible values of *n* in X(k) and $W_{j}\left(k\right)$ signal (where *n* is the dimension of the learning pattern). For this reason, particular components of the *X* pattern are fed sequentially into the network, with all bits of a given component $x_{i}$ provided in parallel. For example, for a ANN processing 10-element learning patterns *X*, only 10 clock cycles are needed. Diagram of the proposed adaptation mechanism is shown in Figure 5.

In the first step, described above, the distances between the neuron weights and the given learning pattern *X* are calculated, according to the so-called Manhattan distance measure:(3)$$D_{L1}\left(X\left(k\right),W_{j}\left(k\right)\right)=\sum_{i=1}^{n}\left|x_{i}\left(k\right)-w_{i,j}\left(k\right)\right|,$$

This means that for each pair ($x_{i}\left(k\right)$, $w_{i,j}\left(k\right)$) the signal $|x_{i}\left(k\right)-w_{i,j}\left(k\right)|$ is calculated, which is then stored in the internal memory of a given neuron. The subtraction operation is based on a multi-bit full subtractor (MBFS), in which the borrow out bit $B_{i,j}$, at the most significant 1-bit full subtractor (1BFS) in the MBFS indicates which of the input signals (*x* or *w*) is larger. This information is also stored in the memory of a given neuron. The memory cells containing particular neuron weights $w_{i,j}$, and the computed $|x_{i}-w_{i,j}|$ factors are indexed with the use of the proposed programmable multiphase clock. It is worth emphasizing here that the described operations are performed in parallel in all neurons of the ANN. In other words, for an *n*-element learning pattern *X*, only *n* clock phases are required to calculate the distances of all neurons in the network to a given *X*. At the next stage of the learning algorithm, the winning neuron is determined. On the basis of the position of this neuron, the ANN then determines which neurons belong to its neighborhood and what is their distance to this neuron. The way the neighborhood is defined depends on the applied learning algorithm of the ANN. Based on the distance, the value of the learning rate, η, is calculated for each neuron in the ANN. Then, based on the previously stored $|x_{i}-w_{i,j}|$ factors, the adaptation process is carried out, as described below:(4)$$\begin{array}{cc} w_{i,j}\left(k+1\right)= & w_{i,j}\left(k\right)+(\bar{B_{i,j}}-B_{i,j})\cdot P_{i,j}\left(k\right)\cdot \left|x_{i}\left(k\right)-w_{i,j}\left(k\right)\right|, \end{array}$$ where $P_{i,j}\left(k\right)$ is the product of learning rate η and the so-called neighborhood function, which determines the strength of the adaptation depending on the distance of a given neuron to the winning neuron. The adaptation process is also carried out in an iterative manner. The number of iterations is equal to the number of iterations from the first step described above (*n*). The same clock generator is used at this stage to control this process. At the hardware level, in Equation (Equation 4) very simple summing, subtraction, negation and multiplication operations are used. Since the $P_{i,j}\left(k\right)$ factor is a fractional numbers, the multiplication is carried out in two steps. First, the $|x_{i}-w_{i,j}|$ is multiplied by the numerator of this number. Since the denominator is one of the powers of the number 2, the division is carried out simply by shifting all bits to the right, by a given number of bits. Such operation is performed by a simple asynchronous circuit composed of a field of switches.

The multi-bit clock generator, described below, is of key importance in the described steps of distance calculation and the adaptation process.

### 2.3. Adaptation Mechanism

The structure of the proposed adaptation mechanism in general corresponds to the adaptation process described above and expressed by Equation (Equation 4). For each clock cycle, following components of the *X* pattern and the vector of neuron weights are provided to the inputs of this block. The final outcome is then computed in an asynchronous fashion. This results from the fact that particular components of this mechanism (Multi-bit full adder (MBFA), Multi-bit full subtractor (MBFS), and multiplier) are asynchronous circuits, leading to a very small delay. This allows obtaining a very high data rate.

A first operation performed by the adaptation system is the subtraction of the *X* and the *W* signal (in following pairs $x_{i}-w_{i,j}$). The result of this operation can be positive or negative. In the second case, if the buffer length is too small (4 bits in this example case), then in two’s complement code the result is incorrect. This problem propagates into the subsequent multiplication operation.

For example, let us consider an example case in which $x_{i}$ = 3 and $w_{i,j}$ = 13. The subtraction result in this case equals −10. In the two’s complement code, assuming the buffer length of 4 bits, we get “0100”. Multiplying this number by η = 8/16 leads (after the shift by 4 bits, required in this case) to the value of “0010”, which corresponds to the number of 2.

To avoid this problem, after the subtraction operation, the absolute value (abs) of the $x_{i}\left(k\right)-w_{i,j}\left(k\right)$ factor is determined. The sign of this factor is stored in a memory cell. As a result, the multiplication may always be performed with positive numbers. The result of the multiplication operation, $P_{i,j}\left(k\right)\cdot \left|x_{i}\left(k\right)-w_{i,j}\left(k\right)\right|$, depending on the stored sign is then changed accordingly. In the case when the sign stored in memory is negative, the sign of the overall result is changed, which is carried out by subtracting the calculated result from the 0 number. In the situation, in which the sign is positive, the result remains unchanged. The EXOR gates used in the circuit are responsible for a proper interpretation of the signs and to ensure a proper operation of the overall adaptation mechanism.

It is worth noting that, for $\left(x_{i}\left(k\right)-w_{i,j}\left(k\right)\right)$ < 0, the EXOR gates behave as typical NOT gates. This allows easily performing either the addition or the subtraction operation, depending on the situation.

The $\eta \left(k\right)\cdot G\left(\right)\cdot (x_{i}\left(k\right)-w_{i,j}\left(k\right))$ signal is, in the last step of the adaptation process, added to a given $w_{i,j}\left(k\right)$ weight of a given neuron. The resultant $w_{i,j}\left(k+1\right)$ substitutes then, in the memory, the previous $w_{i,j}\left(k\right)$ value. Storing the new value into the memory takes place at the falling edge of the clock signal.

Particular $x_{i}$ and $w_{i,j}$ signals are supplied sequentially to the described adaptation block (in each neuron). In this case, the use of a fully parallel approach would be ineffective, as it would require a very large number of external pads of the chip. For example, for ten *x* input signals and 16-bit resolution, it would be necessary to use 160 pads.

#### Programmable Multi-Phase Clock Generator

The proposed multi-phase clock generator has been implemented inside the chip, to reduce the number of external pins of the chip. Circuits of this type are usually implemented using a chain of D-flip flops (DFFs) [27,28,29,30,31,32]. The advantage of such approach is the high immunity of the parameters of the clock to external conditions. However, there are also some disadvantages of such solutions. A typical DFF is composed of 26 transistors (six logic NAND gates). Two of these gates are switched over, even if there is no change at the D input of the DFF. These gates are switches over twice, i.e., during the raising and the falling edges of the controlling clock signal. In a typical multiphase clock generator, the input of only one DFF equals “1”, however the power is dissipated by all DFFs in the chain. As a result, such clocks consume relatively large power.

In this paper, we propose a clock generator that is based on NOT logic gates and switches, connected alternately in series. In this approach, the logical values are stored in parasitic capacitances of particular NOT gates that serve as short-time memory cells. In the realized project, the chain of the switches and NOT gates has been supplemented with additional logic gates that allow determining the clock behavior. As a result, we can select the number of clock phases and the position of the first phase in the clock cycle. Additionally the filling ratio (pulse width) may be determined as well.

The proposed clock generator is shown in Figure 6. It is composed of a chain of 10 blocks denoted as UCLK (unit clock cell). Selected outputs of the UCLK blocks are fed to an nine-input NOR gate. In the situation, in which the outputs of all UCLK block are equal to “0”, the NOR gate generates a new “1” impulse that is provided to the chain, and then propagated between subsequent UCLK blocks, according to subsequent clk1/clk2 pulses. Each UCLK block provides complementary “Nclk*x*” and “Pclk*x*” clock signals, as switches in the ANN are realized as transmission gates, composed of the NMOS and PMOS transistors, connected in parallel.

The structure of the first block in the chain (UCLK_IN) is a bit different than of remaining UCLK blocks, as this circuit plays a different role. It allows providing the generated “1” impulse not necessary to the first UCLK block. For this reason, the output of this block is connected to all UCLK blocks in parallel. The UCLK block that receives the “1” impulse is determined by the sequence of the Bsi9–Bsi0 bits. These bits also determine the number of the clock phases. To explain how it works in detail, let us consider an example sequence of Bsi9–Bsi0 bits of 1011111011. The first appearance of the “0” signal indicates the first phase, while the second occurrence of the “0” signal indicates the last one.

Such a programming method was obtained by an appropriate structure of the UCLK blocks, shown in Figure 7. The advantage of this approach is that reprogramming the clock so that it performs a multi-phase cycle with different numbers of phases, requires changing only 2 or 4 bits, regardless of the maximum number of the phases. In this prototype, the circuit allows obtaining 10 phases, however, by extending the length of the chain of the UCLK blocks and by increasing the number of the inputs of the NOR gate, the clock may be very easily adapted to many phases. Regardless of the length of the chain, reprogramming the clock will always require the same amount of steps.

## 3. Results

In this section, we present selected results obtained during the simulations, as well as the measurements of the prototype chip described in previous sections. We mainly focus on the adaptation mechanism along with the controlling multiphase clock generator—the topic of the presented work. In the case of the adaptation mechanism, we present the simulation results (see Figure 8), while the behavior of the clock generator is demonstrated by means of both the simulations and the laboratory measurements. Main parameters that may be programmed in the case of the clock generator are the number of the clock phases in a given multiphase cycle (see Figure 9), as well as the width of particular clock pulses.

### 3.1. Adaptation Mechanism

Figure 8 illustrates the adaptation process performed by the proposed adaptation mechanism. The top diagram presents subsequent input signals $x_{i}\left(k\right)$ and corresponding neuron weights $w_{i,j}\left(k\right)$. The following three diagrams of Figure 8 show updated weights computed by the proposed circuit. The learning process is presented for different values of the $P_{i,j}\left(k\right)$ factor: 4/16, 8/16, and 14/16. The multiplication by these numbers is performed as a multiplication by the numerator of this number, followed by shifting the resultant product by a given number of bits to the right. It is worth noting that, in the situations when $P_{i,j}\left(k\right)$ equals 4/16 (→ 1/4) or 8/16 (→ 1/2), the multiplication operation may be omitted (1 in numerator). In this case, the $|x_{i}-w_{i,j}|$ term is shifted by different numbers of bits.

Two signals are shown for each simulation presented in Figure 8. The top one is the $\Delta w_{i,j}\left(k\right)=P\left(k\right)\cdot \left[x_{i}\left(k\right)-w_{i,j}\left(k\right)\right]$ signal, which directly results from Equation (Equation 2), while the bottom one is the w(k)±Δw(k), i.e., the weight signal after the adaptation $w_{i,j}\left(k+1\right)$.

### 3.2. Clock Generator

Selected simulation results of the clock generator are shown in Figure 9. Figure 9a illustrates particular input and output signals. Ten top waveforms are subsequent phases generated by the clock. The following two waveforms are two phases of an external clock that controls the designed block. The reset signal, shown below the clock pulses, is used to break a given multiphase cycle. This signal causes that a new cycle starts with the following impulse of the external clock. The bottom waveform is the external “fill” signal that controls the width of all clock phases. Figure 9b,c presents a total supply current, $I_{\mathrm{DD}}$, for the supply voltage of 1.2 V.

Measurements of the clock generator are shown in Figure 10 and Figure 11. Figure 10 illustrates the operation of the clock generator for different numbers of phases: 10, 7 and 3, respectively. The programming sequences in particular cases (bits Bsi9–Bsi0) are as follows: (Figure 10a) “0111111111”; (Figure 10b) “0111111011”; and (Figure 10c) “0110111111”. In all these cases, the first occurrence of the “0” value at first position starts the multiphase cycle from CLK0 phase. The second occurrence of the “0” value stops the cycle, thus starting a new cycle. Case (a) illustrates a full set of clock phases (10 clock phases), thus the second “0” is not required in this case. Figure 11 shows the clock waveforms for different widths, for an example case of five clock phases (programming sequence: “0111101111”).

## 4. Discussion

The presented results show that both the adaptation mechanism and the clock generator work correctly, in line with earlier assumptions. The investigation results, presented in the previous section, show that the realized components of the ANN are flexible solutions. It is especially important in the case of the clock generator, which in this way may be easily adapted to different applications. Here, it serves as one of the key blocks of the ANN realized at the transistor level.

### 4.1. Clock Generator

Since in wireless sensors the power consumption is one of the key parameters, in our designs, we put a special attention to this parameter. Figure 9 shows the supply current waveform over time. The current pulses are generated only during switching the clock to a next phase. In the worst case scenario, when a new “1” impulse is generated by the NOR gate, the value of the current peak equals ca. 175 μA, while an average value during the overall switching period lasting 4 ns does not exceed 50 μA in this case. For the supply voltage of 1.2 V, it means about 240 fJ of energy consumed in this time, in the worst case. In most cases, the consumed energy does not exceed 60–70 fJ per one clock phase.

Another important parameter is the available circuit speed. In the measurements, due to the limitations of the available measurement equipment, the clock was tested up to 20 MHz. In simulations, the achievable speeds were even 150–200 MHz. This, however, requires confirmation by measurements on more advanced measuring equipment.

We compared the results obtained during the tests of the realized clock generator with similar state-of-the art solutions. A comparison is shown in Table 1. To enable a more straightforward comparison, we defined a Figure-of-Merit (FOM), which is the achievable data rate over the power dissipation. Thus, the higher is the value of the FOM, the better is the result. The obtained data rate is relatively small, for example, in the comparison with [32] circuit, however proportionally smaller is also the power dissipation. Thus, the obtained FOM is slightly better than in [32] and substantially better than in other presented cases. It is mostly due to to the substitution of DFFs with dynamic memory cells realized on parasitic capacitances of NOT gates.

### 4.2. Adaptation Mechanism

The possibility of adjusting the width of the clock phase is an important feature, which is useful from the point of view of the adaptation mechanism. This time should be set so that the adaptation system can perform all necessary operations. At the end of a given clock phase (falling edge), a given weight, *i*, in a given neuron, *j*, is updated, which means that $w_{i,j}\left(k\right)$ is substituted with a new signal $w_{i,j}\left(k+1\right)$. For larger signal resolutions, the data processing time is slightly longer.

To facilitate the presentation of the results obtained on the basis of the simulations of the adaptation mechanism, the resolutions of the *x* and the *w* signals have been set to 4. It is sufficient to verify the concept of the circuit. Nevertheless, simulations were carried out also for the resolutions of 8 and 16 bits. Any resolution may be obtained easily by an extension of the lengths of the used summing and subtracting circuits.

In all simulations, to facilitate the interpretation of the results, the equivalent decimal values are also provided, which correspond to their 4-bit counterparts. It should be noted that the Δw(k) term is negative when x<w. In these situations, the 4-bit binary signal is expressed in two’s complement code.

## 5. Conclusions

In this work, we present two circuits, crucial from the point of view of the implementation of artificial neural networks at the transistor level. Such networks can be used in intelligent wireless sensors, which, thanks to the internal processing and analysis of data collected from the environment, will be able to contact the base station less frequently.

Designed circuits—a programmable multi-phase controlling clock and the adaptation block for neuron weights—allows in the next step to build an overall neural network with full learning opportunities at the silicon level. Earlier, as part of our work, we had already designed other components of such networks.

In the paper, we present a prototype specialized chip designed in the CMOS technology. The chip was verified by means of both transistor-level simulations and laboratory measurements. In addition to the described ANN blocks, it also includes other components of wireless sensors, such as an analog-to-digital converter, and a programmable input/output block. Thanks to this, now we can also build an overall prototype intelligent wireless sensor. Remaining components of such sensors, such as a RF communication block, will be based on state-of-the art solutions described in the literature.

## Figures and Tables

**Figure 1 sensors-18-04499-f001:**
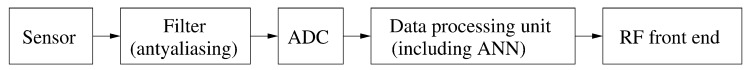
A general structure of the wireless intelligent sensor.

**Figure 2 sensors-18-04499-f002:**
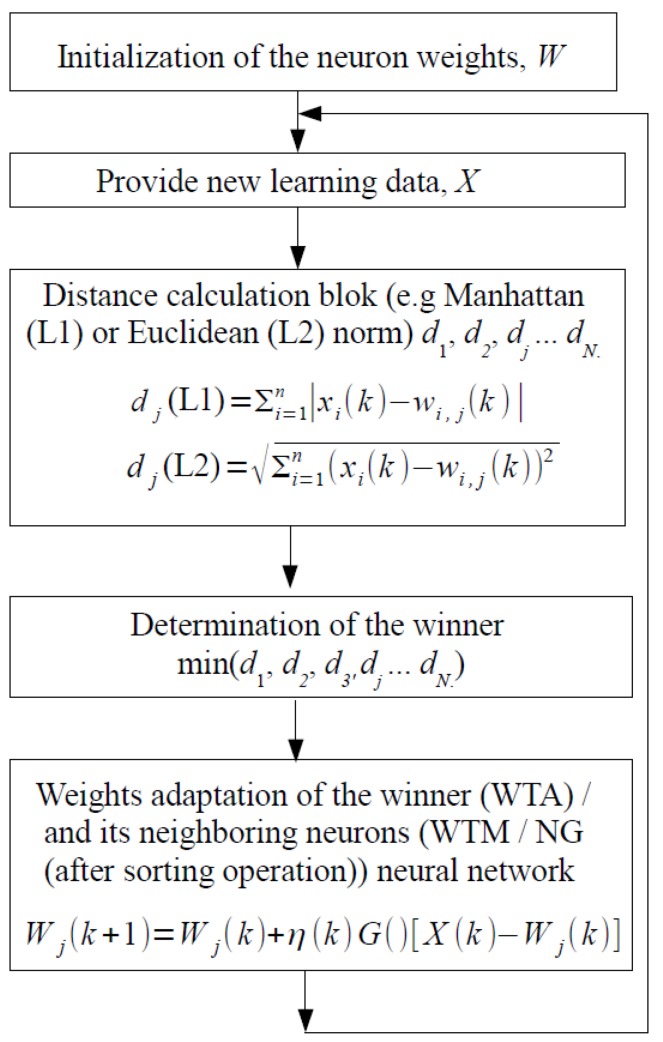
Diagram illustrating following steps of the learning algorithm of typical self-organizing neural networks.

**Figure 3 sensors-18-04499-f003:**
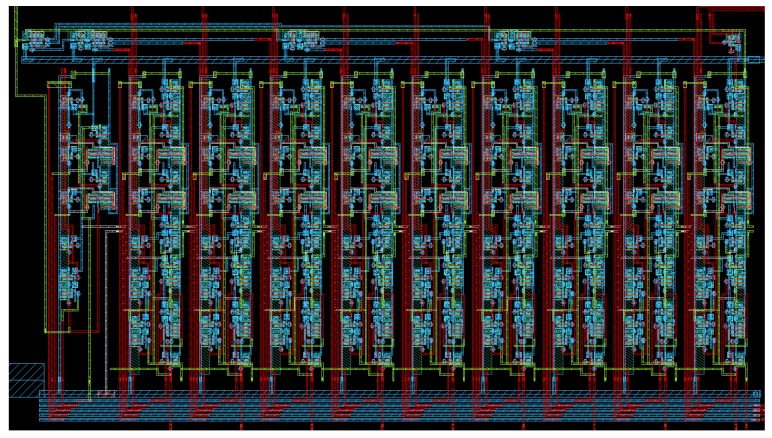
Layout of the proposed, programmable 10-phase clock generator, implemented in the CMOS 130 nm technology. The silicon area equals 0.0058 $\mu^{2}$, with the sizes of 105×55μ.

**Figure 4 sensors-18-04499-f004:**
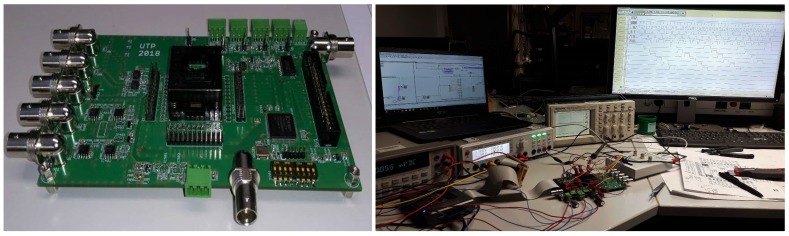
Measurement setup composed of PC board, designed from scratch for the verification of the realized chip prototype: (**left**) a new, printed circuit board (PCB) for the implemented integrated circuit; and (**right**) the overall measurement setup.

**Figure 5 sensors-18-04499-f005:**
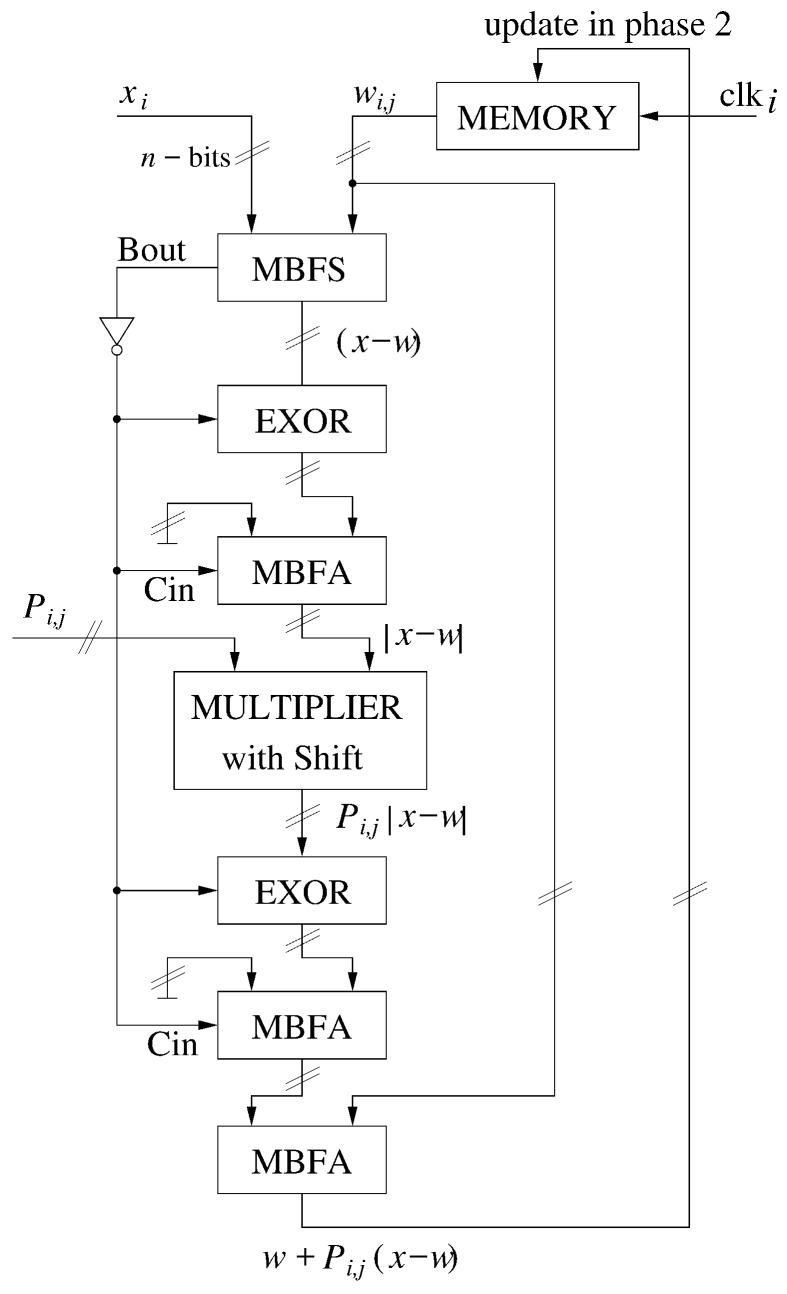
Diagram of the proposed adaptation mechanism.

**Figure 6 sensors-18-04499-f006:**
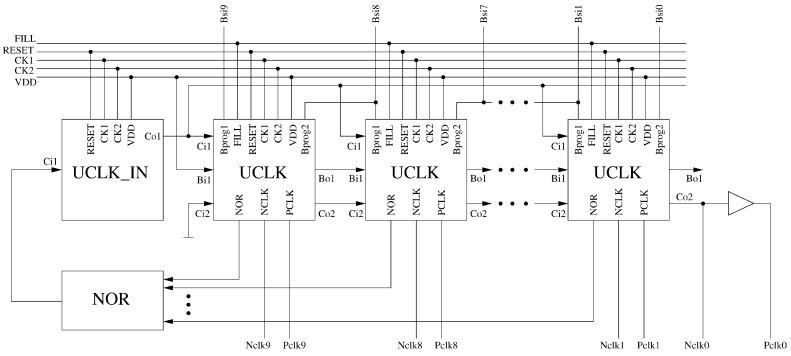
General structure of the proposed 10-phases clock generator.

**Figure 7 sensors-18-04499-f007:**
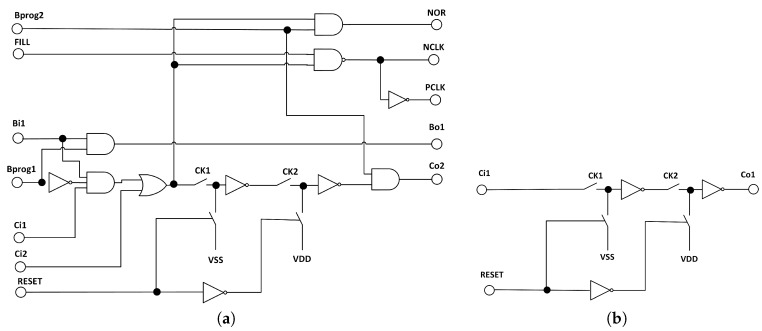
Diagrams of the (**a**) UCLK and (**b**) UCLK_IN sections used in the realized multi-phase clock.

**Figure 8 sensors-18-04499-f008:**
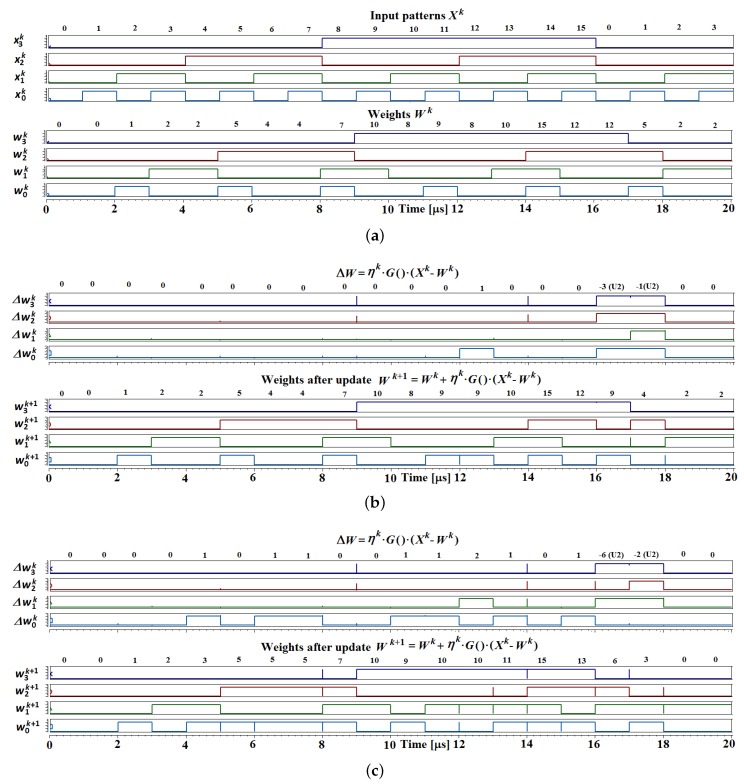
Simulations of the adaptation mechanism for different values of the $P_{i,j}\left(k\right)$ value: (**a**) an example sequence of the input signals $x_{i}\left(k\right)$ and corresponding neuron weights $w_{i,j}\left(k\right)$; (**b**–**d**) values of the updated neuron weights after the adaptation process $w_{i,j}\left(k+1\right)$ for: (**b**) $P_{i,j}\left(k\right)$ = 4/16; (**c**) $P_{i,j}\left(k\right)$ = 8/16; and (**d**) $P_{i,j}\left(k\right)$ = 14/16.

**Figure 9 sensors-18-04499-f009:**
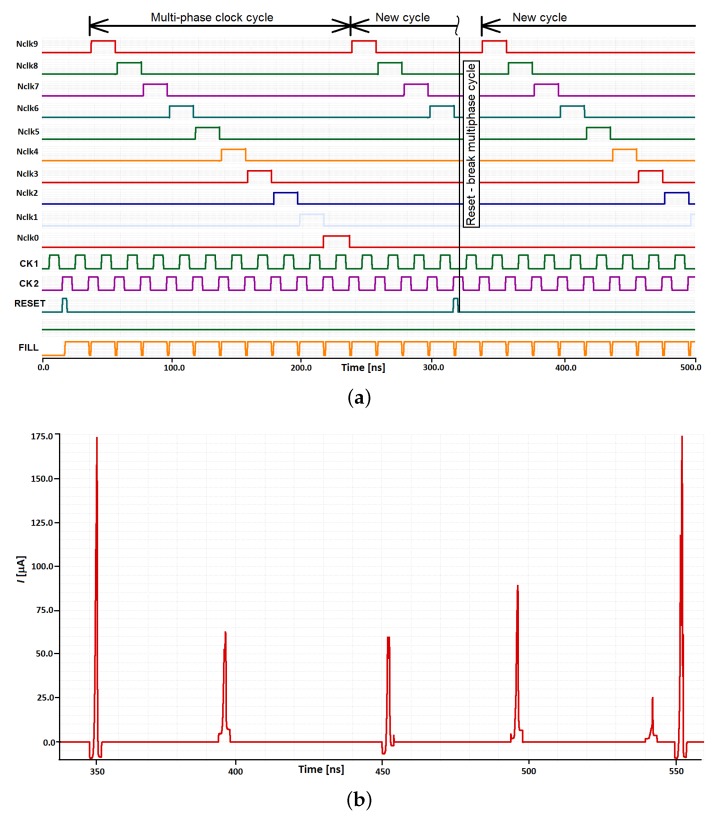
Selected simulation results, illustrating the performance of the designed multi-phase clock, controlled by a simple two-phase external clock: (**a**) selected pulse waveforms for an example case of 10 phases; (**b**) supply current proportional to power dissipation; and (**c**) supply current—the worst case.

**Figure 10 sensors-18-04499-f010:**
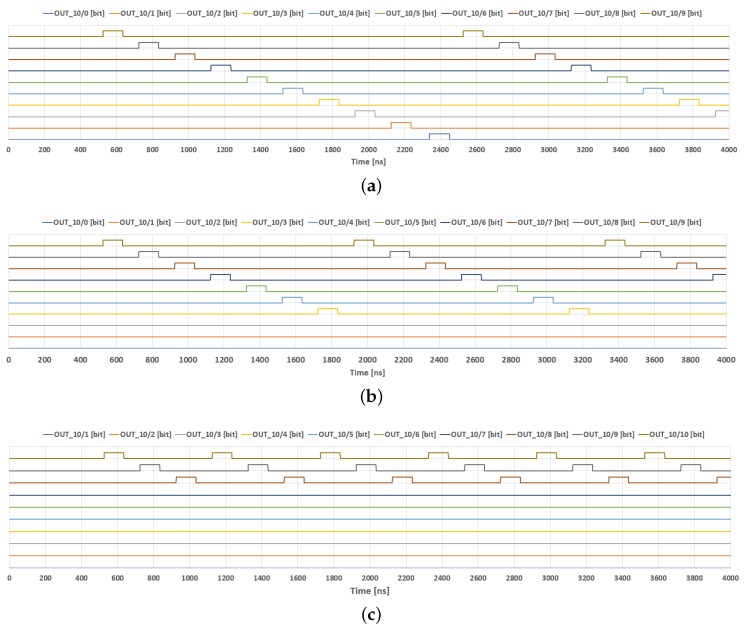
Measurement results of the implemented clock generator for selected numbers of clock phases in a single cycle: (**a**) 10; (**b**) 7; and (**c**) 3.

**Figure 11 sensors-18-04499-f011:**
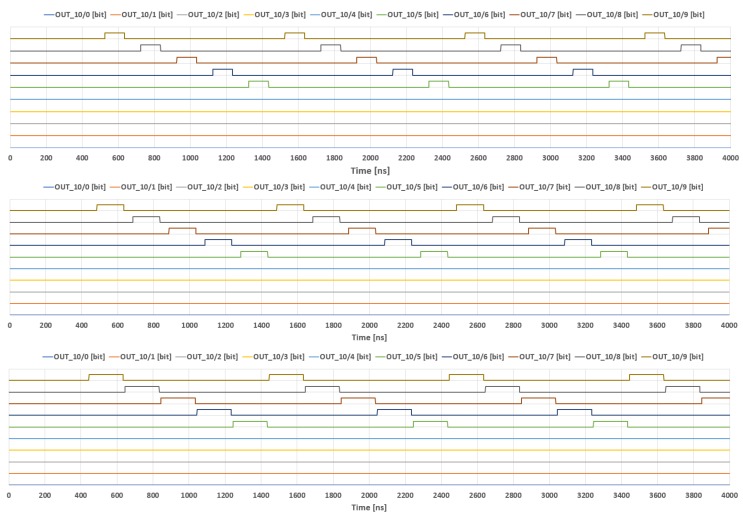
Measurement results of the clock generator with five phases, for different widths of the clock pulses (fill ratio).

**Table 1 sensors-18-04499-t001:** Performance comparisons with other reported clock generators.

Ref.	$f_{\max}\left[\mathrm{GHz}\right]$	P[mW]	$V_{\mathrm{DD}}\left[V\right]$	FOM [1/nJ]	TechnologyCMOS
[30]	1.2	34	1.8	0.035	0.18 μm
[32]	40	45	1.0	0.89	0.09 μm
[33]	0.125	32	1.8	0.004	0.18 μm
[34]	1.8	86.6	3.3	0.021	0.35 μm
[35]	2.0	21	1.2	0.095	0.13 μm
This work	0.15	0.06	1.2	2.5	0.13 μm

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
