# Peer review of "Components of Artificial Neural Networks Realized in CMOS Technology to be Used in Intelligent Sensors in Wireless Sensor Networks"

_sensors, 2018, doi:10.3390/s18124499_

Reviewer 1 Report

This paper presents a novel hardware solution to solve the problem of reduction of the amount of data sent by the sensor to the base station in wireless sensor networks. It proposes a method to miniature integrated artificial neural networks (ANN) applied directly in the sensor can take over the analysis of data collected from the environment, thus reducing amount of data sent over the RF communication block. Meanwhile, a prototype with adaptation mechanism, a programmable multi-phase clock generator and components of the ANN was designed in the CMOS 130nm. At last, paper shows the simulation and measurement results of selected blocks demonstrate the method with good performance.

Comments:

1. In Section 1, paper described the key method is saving energy by reducing the data delivered from sensor to base station. However, the neighbor sensors also need to exchange their data. How to quantify the energy of these two methods? Maybe it is better for authors to cite some results from other papers or present some experiment results.

2. The key algorithm of the adaptation of the weights are WTM、WTA and NG. However, only one paragraph above Section 2 describes them. I think it’s too short to understand for readers. Maybe we can extend some details with one more paragraph or a table about these algorithms.

3. In Figure 2, paper shows two kinds of distance norm, i.e. Euclidean and Manhattan, but in the following parts of paper doesn’t discuss this especially in Simulation and Experiment. Maybe it is better to discuss the feasibility of both distance norm.

4. As we all know, the convergent time of ANN is an important metric to measure the performance of system. Maybe it is better to discuss the convergent time or illustrate by experiment results.

5. Some grammar and spelling errors are still made, such as the word ”wights” in the 6 items below Figure 2, etc. It is better for authors to check throughout whole paper.

Author Response

We are grateful to the reviewers for identifying essential issues and providing us with an important and valuable feedback. We have found the suggestions and comments in the reviews very useful and constructive when preparing a revised version of the manuscript. In what follows, we discuss the details on how the manuscript has been modified to address the concerns raised in the reviews.

Response to Reviewer 1 Comments

Point 1: In Section 1, paper described the key method is saving energy by reducing the data delivered from sensor to base station. However, the neighbor sensors also need to exchange their data. How to quantify the energy of these two methods? Maybe it is better for authors to cite some results from other papers or present some experiment results.

Response 1 :  There are several options possible:

1) The NNs used in particular sensors work only in recall/test phase, which means that all sensors have built-in constant values of the neuron weights.

2) The learning phase of the NNs inside the sensors is allowed.

When the ANN is implemented inside the wireless sensor, then we have n distinct ANNs operating in parallel on a given area, where n is the number of sensors in this area. If data collected by all these sensors need to be processed by an ANN, then if the ANN is implemented in the base station, it has to handle data collected and streamed from all n sensors (a potential  problem with throughput). In this case the ANN would need to be multiplexed and switched between data streams coming from particular sensors. In this situation, in case (1), there is a memory required only to store n computed results for each of the sensors. A larger problem appears in case (2), in which the ANN for each data stream would learn in a bit different way. In this case, it is necessary to store n sets of neuron weights, for each process.

The problem will depend on type of processed data, i.e. the data rate. Former investigations in this area show, that optimized parallel ANN working at data rate of 1 kSamples/s will consume no more than 2-30 uW, assuming 64 - 1000 neurons [23]. Taking the literature studies in the area of wireless data transfer, already presented in the paper, data transfer may take even two or three orders of magnitude larger power.

This issue is discus more detail in the revised version of manuscript. Please refer to page 1 (“An important question here is the one about……”)

Point 2: The key algorithm of the adaptation of the weights are WTM, WTA and NG. However, only one paragraph above Section 2 describes them. I think it’s too short to understand for readers. Maybe we can extend some details with one more paragraph or a table about these algorithms.

Response 2 :  This issue is discus more detail in the revised version of manuscript. Please refer to page 5 (“ Before the learning process, the values of both……”)

Point 3:  In Figure 2, paper shows two kinds of distance norm, i.e. Euclidean and Manhattan, but in the following parts of paper doesn’t discuss this especially in Simulation and Experiment. Maybe it is better to discuss the feasibility of both distance norm.

Response 3 : This issue is discus more detail in the revised version of manuscript. Please refer to page 6 (“One of the important parameters of the presented NNs is the measure……”)

Point 4: As we all know, the convergent time of ANN is an important metric to measure the performance of system. Maybe it is better to discuss the convergent time or illustrate by experiment results.

Response 4 : This issue is discus more detail in the revised version of manuscript. Please refer to page 6 (“ In the literature dealing with the ANNs, an important factor is the learning time”)

Point 5:  Some grammar and spelling errors are still made, such as the word ”wights” in the 6 items below Figure 2, etc. It is better for authors to check throughout whole paper.

Response 5 : The pertinent corrections have been made.

Reviewer 2 Report

Paper “Components of Artificial Neural Networks Realized in CMOS Technology to be used in Intelligent Sensors in Wireless Sensor Networks” presents a prototype integrated circuit, which includes implementations of main components of artificial neural networks (ANN).  Starting from the proposal of the circuit, the authors focus on the adaptation mechanism for the ANN and a multiphase programmable clock used to control such an ANN. The experiments and results on the proposed circuit show that it performs well in line with preconceived assumptions when testing the adaptation mechanism and clock generator.  Thus, I recommend this paper to be accepted after minor revision.

According to the author, the proposed chip is focused on reducing the high energy consumption used in data transfer. However, the author does not refer to the limitations of the chip when incorporating an artificial neural network. That is, limitations in terms of time in processing the data, training the artificial neural network, memory used to store the data, size of training data sets. I mean that if the runtime of the different tasks within the chip with respect to the runtime of the same tasks in a conventional computer compensates for the resolution of the problem of energy consumption used to transfer data if the chip was not implemented by the neural network.

Author Response

We are grateful to the reviewers for identifying essential issues and providing us with an important and valuable feedback. We have found the suggestions and comments in the reviews very useful and constructive when preparing a revised version of the manuscript. In what follows, we discuss the details on how the manuscript has been modified to address the concerns raised in the reviews.

Response to Reviewer 2 Comments

Point 1: According to the author, the proposed chip is focused on reducing the high energy consumption used in data transfer. However, the author does not refer to the limitations of the chip when incorporating an artificial neural network. That is, limitations in terms of time in processing the data, training the artificial neural network, memory used to store the data, size of training data sets. I mean that if the runtime of the different tasks within the chip with respect to the runtime of the same tasks in a conventional computer compensates for the resolution of the problem of energy consumption used to transfer data if the chip was not implemented by the neural network.

Response 1:  At this problem we should look not only from the point of view of the energy consumption itself, although this aspect is one of the most important ones. From the point of view of the sensor itself, if data processing is performed on-board, and then the communication is say 100 times less frequent, it is energy more profitable. This is shown by previous studies in this topic, e.g. [1,2]. The solution to some extent depends also on the type of the processed data, i.e. the data rate. Former investigations in this area [23] show, that optimized parallel ANN working at data rate of 1 kSamples/s may consume about 2-30 uW, for 64 - 1000 neurons, respectively. Other works cited in the paper, in the area of wireless data transfer, show that data transfer may take even two or three orders of magnitude larger power (c. 15 mW). 

Taking into account other aspects, in general there are two options possible:

1) The ANNs used in particular wireless sensors work only in the recall/test phase. This means that all sensors are programmed and have built-in constant values of particular neuron weights.

2) The adaptation of neuron weights (learning phase) inside the sensors is allowed.

When the ANN is implemented inside the wireless sensor, then we have N distinct ANNs operating in parallel on a given area, in which the wireless sensor network operates. Here N is the number of sensors in this area. If data collected by all these sensors need to be processed by an ANN, then if the ANN is implemented in the base station, it has to handle data collected and streamed from all the N sensors (a potential throughput issue). In this case, the ANN may need to be multiplexed and switched between data streams coming from particular sensors (additional problem appears when there is no synchronization of the sensors). In this situation, in case (1), there is a memory required only to store N computed results for each of the sensors. A larger problem appears in case (2), in which the ANN for each data stream would learn in a bit different way and thus need to be switched between values of particular parameters of particular ANNs. In this case, for example, it is necessary to store N sets of neuron weights, for each process.

This issue is discus more detail in the revised version of manuscript. Please refer to page 1 (“An important question here is the one about……”)

Round  2

Reviewer 1 Report

The authors have addressed all my concerns. I recommend to accept this paper.